# Two-Stage Calibration Scheme for Magnetic Measurement System on Guided Munition

**DOI:** 10.3390/s21175799

**Published:** 2021-08-28

**Authors:** Yuyang Xue, Xiaoming Zhang

**Affiliations:** 1National Key Laboratory for Electronic Measurement Technology, North University of China, Taiyuan 030051, China; xyy_nav@163.com; 2Key Laboratory of Instrumentation Science & Dynamic Measurement of Ministry of Education, North University of China, Taiyuan 030051, China

**Keywords:** magnetic measurement system, misalignment angles, three-position calibration, ellipsoid fitting

## Abstract

In order to calibrate the magnetic measurement system used in guided munition on site, a two-stage calibration (TSC) scheme without reference is proposed in this paper. Analyzing the interfering magnetic field in the projectile and misalignment angles between the projectile coordinate system and measurement coordinate system establishes a proper mathematical equivalent model and derives a calibration method. The first stage is ellipsoid fitting to obtain the equivalent zero-offset, equivalent sensitivity and equivalent non-orthogonal angles of the sensor; the second stage is to calibrate the misalignment angles between the projectile coordinate system and the measurement coordinate system with the three-position calibration (TPC) method. Complete calibration is convenient to operate and does not need an additional reference, which has wide applicability. The simulation results show that the deviation in the measured value after compensation is within 100 nT. The experiment proves that the error of compensated magnetic value is about 150 nT, which meets the accuracy of requirements in guided munitions.

## 1. Introduction

In the field of navigation technology, the triaxial magnetic measurement system, with the AMR magnetometer as the core sensor which measures the earth’s magnetic field to obtain precise directional information [1], has been widely applied to vehicle detection [2], resource exploration [3], magnetic anomaly detection and navigation [4,5,6], due to the advantages of using a magnetometer such as small space occupation, fast response speed, good anti-overload performance, low cost, etc. [4,7]; as well, it is used in guided munitions. Generally, the magnetic measurement system must be calibrated in advance [8]; many methods need a high-precision turntable or an additional reference, however, in the field of guided munition, due to the long-term manufacturing, storage, transportation and assembling, the interfering magnetism of the projectile will change, meaning that the parameters are inaccurate which are calibrated before, such as zero-offset, sensitivity and non-orthogonal angles [9]. Besides, there are misalignment angles between the projectile coordinate system and measurement coordinate system [10]. Theoretically, a one-degree misalignment angle results in thousands of nT errors of measured value [11], which need to rotate the sensor coordinate system to coincide with the projectile coordinate system.

Aiming towards the calibration for the magnetometer and the magnetic measurement system, some scholars has proposed some effective methods. Valérie Renaudin proposed a complete calibration in magnetic domain in [11], they analyzed the reasons of errors and fitted the measured value of magnetometer as a sphere; however, the equation represents a second-order surface: it may be a hyperboloid, a cone surface or an ellipsoid, there is no specific restriction, and it cannot judge the validity of the calibration after the process is completed. Yanxia Liu proposed a two-stage calibration based on particle swarm optimization in [12]. Particle swarms belong to a stochastic global optimal algorithm, and can calibrate all parameters of sensors and improve the accuracy of triaxial sensor. Hard magnetism and soft magnetism were considered when they established the error model; however, some necessary components are ignored in the analysis, which cannot be estimated and extra deviations still exist, and the particle swarm optimization algorithm has its disadvantages: the more complex the surrounding environment is, the more samples are needed to describe the posterior probability distribution, and the higher the complexity of the algorithm, the less it is suitable for calibrating magnetic measurement systems on guided munitions on site. Supeng Li proposed a method to calibrate the misalignment angle in [13]; the method is based on the principle that the trajectory of the measurement data analyzes the graphical feature to obtain the constrained relationship among the components of the coordinate system. A three-step calibration method for tri-axial sensors was proposed in [14]; the method aims to calibrate the magnetometer and accelerometer simultaneously, one of the two kinds of sensors is calibrated first to define an internal reference, which is not suitable for measurement systems with one sensor. Donghui Liu proposed a simplified ellipsoid fitting to calibrate magnetometers [15]; however, they could not solve the problem of misalignment angles. Lee Jung analyzed the misalignment and soft-iron distortion to calibrate magnetometers, however this is the internal misalignment error of the sensor [16]. Tandon P. proposed a method based on ellipsoid fitting [17]; ellipsoid fitting is a classical method to calibrate field vector sensors, however, projectiles cannot be installed on the calibration equipment. The methods above can improve the accuracy of sensors or systems effectively, however, some of them need additional high-precision equipment or an external reference, and cannot calibrate the magnetic measurement system on the projectile separately. Some scholars proposed some methods with EKF [18,19,20,21] or UKF [22,23,24] to estimate measured value, however, the nonlinear model has errors in linearization, and magnetic measurement systems are affected by the hard magnetism and soft magnetism; they are not applicable for magnetic measurement systems, and the errors need to be eliminated effectively in advance. Yanke Wang proposed a calibration method of magnetometer based on a BP neural network [25]; to improve the training speed and convergence of neural networks, the Levenberg Marquardt backpropagation training algorithm is designed to calibrate the magnetometer. This method can effectively reduce the error caused by the change in magnetometer parameters; the measurement error of a magnetometer can be less than 10 nT, however, the influence of misalignment errors is not considered in their model. Further, the prediction ability of the neural network is related to the selected training model; when predicting different measurement systems, the surrounding environment is different, and the required training models are also different. In addition, the BP neural network method has an over-fitting phenomenon, and the limit position of over-fitting is also difficult to determine to ensure the optimal prediction.

Therefore, aiming at the various errors of measurement systems and inconvenient conditions, this paper proposes a complete calibration scheme without additional reference and uses ellipsoid fitting based on the Least Square method and TPC method to obtain equivalent parameters and misalignment angles to finish fast calibration for measurement systems on site.

## 2. Model of Measurement System

Although micro-mini magnetic sensors have different sensitivity mechanisms to magnetic fields and different sensing characteristics, manufacturing errors are related to various factors. For the convenience of analysis, considering the relationship of the input–output of the magnetic sensor, the errors on the measurement system mainly include two types: single-axis sensing error and three-axis coupling error [26].

### 2.1. Error of Single-Axis Sensor

The mathematical model of the single-axis magnetic sensor can be expressed as:(1)$$B_{i}^{s}=(k_{n,i}+\delta k_{i})B_{e,i}^{s}+B_{0,i}^{s}=k_{i}B_{e,i}^{s}+B_{0,i}^{s}$$
where, *k_n_*_,*i*_ (*i* = x, y, z) is standard sensitivity, *δk_i_* (*i* = x, y, z) is deviation of sensitivity, *k_i_* (*i* = x, y, z) is real sensitivity and *B^s^*_0,*i*_ (*i* = x, y, z) is zero offset.

### 2.2. Error of Triaxial Sensor

The errors of the triaxial magnetic sensor are mainly the mismatch error of zero-offset among the three axes, the static sensitivity mismatches error and the non-orthogonal error among the three axes [27]. The mathematical model of zero-offset and sensitivity can be expressed as:(2)$$\left\{ \begin{array}{l} H^{s}=K_{M,sensitivity}H_{e}^{s}\\ K_{M,sensitivity}=k_{M,n}I+diag[\delta k_{M,x},\delta k_{M,y},\delta k_{M,z}] \end{array}\right.$$
where *k_M_*_,_*_n_* is the theoretical sensitivity of the sensor and *k_M_*_,_*_x_*, *k_M_*_,*y*_ and *k_M_*_,*z*_ are real sensitivities of three axes respectively. Ideally, *k_M_*_,_*_n_* = 1.

The non-orthogonal error among the axes is the measurement error caused by the fabrication that the three sensitive axes cannot be orthogonal completely to each other during manufacturing [28]. For the convenience of analysis, determine the ideal coordinate system as ‘s′’ system, determine the real coordinate system as ‘s’ system; *Ox^s^*^’^ coincides with *Ox**^s^*, illustrated in Figure 1.

Considering that the non-orthogonal angles *δ_M_*_,1_*, δ_M_*_,2_ and *δ_M_*_,3_ are all small angles, the non-orthogonal angle matrix *C_M,nonorth_* can be simplified as:(3)$$C_{M,nonorth}\approx \left[\begin{array}{ccc} 1 & -\delta_{M,1} & -\delta_{M,2}\\ 0 & 1 & -\delta_{M,2}\delta_{M,3}\\ 0 & 0 & 1 \end{array}\right]$$

The mechanical misalignment angles of the magnetic sensor mainly consist of welding alignment errors and assembling alignment errors.

### 2.3. Misalignment Angles

After orthogonal calibration, the measurement coordinate system and the projectile coordinate system are both the orthogonal coordinate system. The conversion relationship between the two can be regarded as the rotation transformation between the two orthogonal coordinate systems, described by Euler angles. The rotation sequence of the two coordinate systems is defined as that the projectile coordinate system first rotates *α_y_* around the Y-axis, then rotates *α_z_* around the Z-axis and finally rotates *α_x_* around the X-axis, then the projectile coordinate system coincides with the orthogonal magnetic sensitive coordinate system, as shown in Figure 2. The conversion matrix is defined as *C_b_^s^*^’^.
(4)$$\begin{array}{l} C_{b}^{s^{\prime }}=R_{x}(\alpha_{x})R_{z}(\alpha_{z})R_{y}(\alpha_{y})\\ =\left[\begin{array}{ccc} \cos\alpha_{y}\cos\alpha_{z} & \sin\alpha_{z} & -\sin\alpha_{y}\cos\alpha_{z}\\ \sin\alpha_{x}\sin\alpha_{y}-\cos\alpha_{x}\cos\alpha_{y}\sin\alpha_{z} & \cos\alpha_{x}\cos\alpha_{z} & \sin\alpha_{x}\cos\alpha_{y}+\cos\alpha_{x}\sin\alpha_{y}\sin\alpha_{z}\\ \cos\alpha_{x}\sin\alpha_{y}+\sin\alpha_{x}\cos\alpha_{y}\sin\alpha_{z} & -\sin\alpha_{x}\cos\alpha_{z} & \cos\alpha_{x}\cos\alpha_{y}-\sin\alpha_{x}\sin\alpha_{y}\sin\alpha_{z} \end{array}\right] \end{array}$$

### 2.4. Disturbing of Magnetic Field

#### 2.4.1. Hard Magnetism

The magnetic field is generated by the hard magnetic materials in the projectile [29]. Hard magnetic materials have high coercivity and remanence value. Once magnetized by an external magnetic field, the remanence can be retained for a long time and will not disappear easily. The projectile body is mainly made of some ferromagnetic materials such as steel alloy; the shells of much of the electromechanical equipment on the projectile are also made of ferromagnetic materials. During the manufacture or storage of the projectile, it is parked at a fixed position for a long time, and it is continuously magnetized by the geomagnetic field in that direction at that location, so that the hard magnetic material in the projectile is magnetized and has strong magnetism. When the sensor on the projectile, the error of hard magnetic materials can be equivalent to a bias, which can be shown as:(5)$$B_{p}^{b}={\left[\begin{array}{ccc} B_{p,x}^{b} & B_{p,y}^{b} & B_{p,z}^{b} \end{array}\right]}^{T}$$

#### 2.4.2. Soft Magnetism

Soft magnetism is produced by the soft magnetic materials in the equipment. For the convenience of analysis, the soft magnetic material in the projectile is decomposed into countless, infinitely thin soft magnetic wires parallel to the axial, vertical and transverse directions of the projectile coordinate system equivalently. The axial, vertical and transverse soft magnetic wires are affected by the geomagnetic field, respectively. The axial soft magnetic filament clusters are magnetized by the *O_x_^b^* axial component $B_{G,x}^{b}$ of the geomagnetic field in the equipment coordinate system, and the inductive magnetic field is proportional to $B_{G,x}^{b}$, $\left\|B_{xi}^{b}\right\|=lB_{G,x}^{b}$ (*l* is a proportional coefficient). The inductive magnetic field vector of the axial soft magnetic filament cluster is shown in Figure 3, which is decomposed by projection on the coordinate axis of the projectile coordinate system *Ox^b^*, *O_y_^b^* and *O_z_^b^*, respectively, and $H_{xi,x}^{b}=c_{11}H_{G,x}^{b}$, $H_{xi,y}^{b}=c_{21}H_{G,x}^{b}$ and $H_{xi,z}^{b}=c_{31}H_{G,x}^{b}$.

In the same way, the vertical soft magnetic filament clusters and transverse soft magnetic filament clusters of the projectile are magnetized by the geomagnetic field components $B_{G,y}^{b}$ and $B_{G,z}^{b}$, respectively, and all the vertical soft magnetic filament clusters and transverse soft magnetic filament clusters on the projectile are correspondingly magnetized. The magnitudes of the induced magnetic field before and after magnetization of the geomagnetic component are $\left\|B_{yi}^{b}\right\|=mB_{G,y}^{b}$ and $\left\|B_{zi}^{b}\right\|=nB_{G,z}^{b}$, respectively, where *m* and *n* are the corresponding proportional coefficients and their orientations are related to the installation position and attitude of the magnetic sensor in the equipment. Project $B_{yi}^{b}$ and $B_{zi}^{b}$ to *O_x_^b^*, *O_y_^b^*, *O_z_^b^* in the projectile coordinate system can be shown as:(6)$$\begin{array}{l} B_{i}^{b}=C_{induce}B_{G}^{b}\\ C_{induce}=\left[\begin{array}{ccc} c_{11} & c_{12} & c_{13}\\ c_{21} & c_{22} & c_{23}\\ c_{31} & c_{32} & c_{33} \end{array}\right] \end{array}$$

#### 2.4.3. Eddy Current Magnetic Field

The eddy current magnetic field is excited by the electric eddy current to generate a corresponding inductive magnetic field around it. The eddy current magnetic field is related to factors such as the rate of change of the environmental magnetic field, the motion parameters of the conductor, the geometry of the conductor, the permeability and the conductivity of the conductor. During the external ballistic flight of the projectile, the eddy current magnetic field of the projectile is mainly caused by a series of closed loops in the metal of the projectile shell when the projectile performs high-speed axial translation, rotation around the axis or maneuvering in the outer ballistic flight. The magnetic flux passing through the enclosed area of the closed loop changes with the changes of the three-axis components of the geomagnetic field in the projectile coordinate system, and then generates eddy currents in the loop. This electric eddy current generates an eddy current magnetic field to prevent this change [30,31].

According to the theory about electromagnetic field, the influence of the eddy current magnetic field on the projectile magnetic sensor can be described as:(7)$$\begin{array}{l} B_{eddy}^{b}=E_{eddy}\frac{dB_{G}^{b}}{dt}\\ E_{eddy}=\left[\begin{array}{ccc} e_{11} & e_{12} & e_{13}\\ e_{21} & e_{22} & e_{23}\\ e_{31} & e_{32} & e_{33} \end{array}\right] \end{array}$$

The changes in the projectile are mainly caused by the steering gear, the rudder blades and the attitude adjustment engine, etc. [32]. For the convenience of analysis, the eddy current magnetic field of the projectile can be described as a three-dimensional random vector, that is:(8)$$B_{n}^{b}={\left[\begin{array}{ccc} B_{n,x}^{b} & B_{n,y}^{b} & B_{n,z}^{b} \end{array}\right]}^{T}$$

It meets the following conditions:$$\begin{array}{l} E\left[\Delta B_{n}^{b}\right]=0_{3\times 1};\\ E\left[\left(\Delta B_{n}^{b}\right){\left(\Delta B_{n}^{b}\right)}^{T}\right]=diag\left[\begin{array}{ccc} \sigma_{H_{n,x}^{b}}^{2} & \sigma_{H_{n,y}^{b}}^{2} & \sigma_{H_{n,z}^{b}}^{2} \end{array}\right] \end{array}$$

Therefore, the actual measured magnetic field value of the projectile can be expressed as:(9)$$\begin{array}{ll} B_{e}^{b} & =B_{G}^{b}+B_{distrub}^{b}\\ & =B_{G}^{b}+\left(B_{p}^{b}+C_{induce}H_{G}^{b}+E_{eddy}\frac{dB_{G}^{b}}{dt}+{\bar{B}}_{n}^{b}+\Delta B_{n}^{b}\right)\\ & =\left(I_{3\times 3}+C_{induce}\right)B_{G}^{b}+E_{eddy}\frac{dB_{G}^{b}}{dt}+\left(B_{p}^{b}+{\bar{B}}_{n}^{b}\right)+\Delta B_{n}^{b} \end{array}$$

Substituting it into the magnetic sensor measurement model, the mathematical model of the measurement information of the projectile can be obtained as:(10)$$\begin{array}{ll} B^{s} & =K_{M,sensitivity}C_{M,nonorth}C_{b}^{s^{\prime }}B_{e}^{b}+B_{0}^{s}+B_{n}^{s}\\ & =K_{M,sensitivity}C_{M,nonorth}C_{b}^{s^{\prime }}\left(B_{G}^{b}+B_{distrub}^{b}\right)+B_{0}^{s}+B_{n}^{s}\\ & =K_{M,sensitivity}C_{M,nonorth}C_{b}^{s^{\prime }}\left[\left(I_{3\times 3}+C_{induce}\right)B_{G}^{b}+E_{eddy}\frac{dB_{G}^{b}}{dt}+\left(B_{p}^{b}+{\bar{B}}_{n}^{b}\right)+\Delta B_{n}^{b}\right]+B_{0}^{s}+B_{n}^{s} \end{array}$$

Then, the relationship of input and output of the magnetic sensor can be simplified as:(11)$$B^{s}=K_{M}B_{G}^{b}+K_{eddy}\frac{dB_{G}^{b}}{dt}+B_{offset}^{s}+B_{noise}^{s}$$
where *K_M_* is an equivalent sensitivity matrix, *K_eddy_* is an equivalent eddy current sensitivity coefficient matrix, ***B^s^_offset_*** is an equivalent zero-offset and ***B^s^_noise_*** is an equivalent noise in output whose mean value is zero.

If the influence of the eddy current magnetic field of the projectile is ignored, it can be approximated as a linear model:(12)$$B^{s}=K_{M}B_{G}^{b}+B_{offset}^{s}+B_{noise}^{s}$$
where:


$\begin{array}{l} K_{M}=K_{M,sensitivity}C_{M,nonorth}C_{b}^{s^{\prime }}\left(I_{3\times 3}+C_{induce}\right);\\ B_{offset}^{s}=K_{M,sensitivity}C_{M,nonorth}C_{b}^{s^{\prime }}\left(B_{p}^{b}+{\bar{B}}_{n}^{b}\right)+B_{0}^{s}\\ B_{noise}^{s}=K_{M,sensitivity}C_{M,nonorth}C_{b}^{s^{\prime }}\Delta B_{n}^{b}+B_{n}^{s} \end{array}$


The parameters (equivalent sensitivity coefficient matrix *K_M_*, equivalent zero-offset ***B^s^_offset_***) in the output information of the magnetic sensor are obtained by the first stage calibration method, and the geomagnetic information can be estimated by the following formula:(13)$$B_{G}^{b}=K_{M}^{-1}(B^{s}-B_{offset}^{s})$$

## 3. Calibration Scheme

According to the analysis above, the calibration process can be decomposed into two stages: calibration of sensor and calibration of measurement system:(1)The first stage is to finish the unitization and orthogonalization of the coordinate system of the sensor;(2)The second stage is to calibrate misalignment angles between the measurement system and projectile coordinate system by the TPC method based on the first stage.

### 3.1. First-Stage Calibration

In the calibration area where the geomagnetic field is stable, the geomagnetic information is measured by an ideal orthogonal magnetometer that does not contain various measurement error factors. When rotating the sensor in space, the measured value can be described as a sphere [33]. Then, the measured value of the ideal magnetic sensor in different attitudes satisfies the following equation:(14)$$\begin{array}{ll} {\left\|B_{G}^{b}\right\|}^{2} & ={\left(B_{G}^{b}\right)}^{T}\cdot B_{G}^{b}={\left(C_{c}^{b}B_{G}^{c}\right)}^{T}\cdot \left(C_{c}^{b}B_{G}^{c}\right)\\ & ={\left(B_{G}^{c}\right)}^{T}{\left(C_{c}^{b}\right)}^{T}\left(C_{c}^{b}\right)\left(B_{G}^{c}\right)\\ & ={\left(B_{G}^{c}\right)}^{T}\left(B_{G}^{c}\right)={\left\|B_{G}^{c}\right\|}^{2}\\ & =const \end{array}$$

According to the mathematical model mentioned before in the article, the measured value of the geomagnetic field is:(15)$$B_{G}^{b}=K_{M}^{-1}\left(B^{s}-B_{offset}^{s}\right)=C_{s^{\prime }}^{b}C_{s}^{s^{\prime }}\left(B^{s}-B_{offset}^{s}\right)$$

Therefore, there is:(16)$$\begin{array}{ll} {\left\|B_{G}^{b}\right\|}^{2} & ={\left(B_{G}^{b}\right)}^{T}B_{G}^{b}={\left(B^{s}-B_{offset}^{s}\right)}^{T}{\left(K_{M}^{-1}\right)}^{T}\left(K_{M}^{-1}\right)\left(B^{s}-B_{offset}^{s}\right)\\ & ={\left(B^{s}-B_{offset}^{s}\right)}^{T}{\left(C_{s}^{s^{\prime }}\right)}^{T}{\left(C_{s^{\prime }}^{b}\right)}^{T}\left(C_{s^{\prime }}^{b}\right)\left(C_{s}^{s^{\prime }}\right)\left(B^{s}-B_{offset}^{s}\right)\\ & ={\left(B^{s}-B_{offset}^{s}\right)}^{T}{\left(C_{s}^{s^{\prime }}\right)}^{T}\left(C_{s}^{s^{\prime }}\right)\left(B^{s}-B_{offset}^{s}\right) \end{array}$$

After simplification, it satisfies the quadratic standard equation:(17)$${\left(B^{s}\right)}^{T}\frac{{\left(C_{s^{\prime }}^{s}\right)}^{T}\left(C_{s^{\prime }}^{s}\right)}{{\left\|B_{G}^{b}\right\|}^{2}}B^{s}-2{\left(B^{s}\right)}^{T}\frac{{\left(C_{s}^{s^{\prime }}\right)}^{T}\left(C_{s}^{s^{\prime }}\right)}{{\left\|B_{G}^{b}\right\|}^{2}}B_{offset}^{s}+{\left(B_{offset}^{s}\right)}^{T}\frac{{\left(C_{s}^{s^{\prime }}\right)}^{T}\left(C_{s}^{s^{\prime }}\right)}{{\left\|B_{G}^{b}\right\|}^{2}}\left(B_{offset}^{s}\right)=1$$

Then, the measurement data satisfies a quadratic ellipsoid surface equation, and its geometric meaning is: the three-dimensional coordinates determined by the measurement data are on the three-dimensional ellipsoid surface, and the center of the ellipsoid is equivalent zero-offset. The radius of each semi-axis of the ellipsoid is related to the product of the equivalent sensitivity coefficient of the corresponding axis of the magnetic sensor and the modulus of the geomagnetic field. The angle between each semi-axis of the ellipsoid and the sensitive axis is the equivalent non-orthogonal angle between each axis of the magnetic sensor.

Suppose the equation of the ellipsoid as the following equation:(18)$$F(\xi ,z)=\xi^{T}z=ax^{2}+by^{2}+cz^{2}+2dxy+2exz+2fyz+2px+2qy+2rz+g=0$$
where $\xi ={\left[a,b,c,d,e,f,p,q,r,g\right]}^{T}$ are the coefficients that need to be calculated and $z={\left[x_{i}^{2},y_{i}^{2},z_{i}^{2},2x_{i}y_{i},2x_{i}z_{i},2y_{i}z_{i},2x_{i},2y_{i},2z_{i},1\right]}^{T}$ are the measured values of the tri-axial measurement vector. Fitting measured data to obtain the optimal ellipsoid fitting coefficients ${\left[a,b,c,d,e,f,p,q,r,g\right]}^{T}$, in the process of fitting the ellipsoid surface, the minimum square sum of the algebraic distance between the measurement data and the ellipsoid surface is determined as the criterion as:(19)$$\underset{\xi \in R^{6}}{\min}{\left\|F(\xi ,m_{i})\right\|}^{2}=\underset{\xi \in R^{6}}{\min}\xi^{T}D^{T}D\xi$$
where:$$D=\left[\begin{array}{cccccccccc} x_{1}^{2} & y_{1}^{2} & z_{1}^{2} & 2x_{1}y_{1} & 2x_{1}z_{1} & 2y_{1}z_{1} & 2x_{1} & 2y_{1} & 2z_{1} & 1\\ x_{2}^{2} & y_{2}^{2} & z_{2}^{2} & 2x_{2}y_{2} & 2x_{2}z_{2} & 2y_{2}z_{2} & 2x_{2} & 2y_{2} & 2z_{2} & 1\\ \vdots & \vdots & \vdots & \vdots & \vdots & \vdots & \vdots & \vdots & \vdots & \vdots \\ x_{N}^{2} & y_{N}^{2} & z_{N}^{2} & 2x_{N}y_{N} & 2x_{N}z_{N} & 2y_{N}z_{N} & 2x_{N} & 2y_{N} & 2z_{N} & 1 \end{array}\right]$$[*x_i_, y_i_, z_i_*]^T^ (*i =* 1, 2, …… *N*) is the measurement data of the projectile in the *i*-th rotation attitude. Calculating with vector (*X* − *X*_0_)*^T^ A* (*X* − *X*_0_) = 1, then
(20)$$X^{T}AX-2X_{0}^{T}AX+X_{0}^{T}X_{0}=1$$
where $A=\left[\begin{array}{ccc} a & d & e\\ d & b & f\\ e & f & c \end{array}\right]$ is the matrix related to the semi-axis of the ellipsoid and its rotation angle and $X_{0}=-A^{-1}\left[\begin{array}{c} p\\ q\\ r \end{array}\right]$ is the center point coordinates of the ellipsoid.
(21)$$\left\{ \begin{array}{l} \frac{{\left(C_{s}^{s^{\prime }}\right)}^{T}\left(C_{s}^{s^{\prime }}\right)}{{\left\|B_{G}^{b}\right\|}^{2}}=A\\ B_{offset}^{s}=X_{0} \end{array}\right.$$

Determine that $P={\left(C_{s}^{s^{\prime }}\right)}^{T}\left(C_{s}^{s^{\prime }}\right)={\left\|B_{G}^{b}\right\|}^{2}A$, *P* is a real symmetric positive definite matrix. According to the Cholesky decomposition theorem: when *P* is a real symmetric positive definite matrix of order *n*, then *P* has a unique *LDU* decomposition, shown as:(22)P=LDU
where *D* is the diagonal matrix *D* = *diag* (*d*1, *d*2, …, *dn*) and *di* > 0 (*i* = 1, 2, … *n*); *L* is the unit lower triangular matrix; *U* is the unit upper triangular matrix.

Determine $\tilde{D}=diag(\sqrt{d_{1}},\sqrt{d_{2}},\cdots ,\sqrt{d_{n}})$, then
(23)$$P=L{\tilde{D}}^{2}U$$

Due to *P^T^ = P*, then
(24)$$U^{T}{\tilde{D}}^{2}L^{T}=L{\tilde{D}}^{2}U$$

According to the uniqueness of the matrix Cholesky decomposition, $U^{T}=L;L^{T}=U$.
(25)$$\begin{array}{ll} P & =L{\tilde{D}}^{2}U=U^{T}{\tilde{D}}^{2}U\\ & =U^{T}{\tilde{D}}^{T}\tilde{D}U={\left(\tilde{D}U\right)}^{T}\left(\tilde{D}U\right) \end{array}$$

The matrix $C_{s}^{s^{\prime }}$ in corresponding model is:(26)$$C_{s}^{s^{\prime }}=\tilde{D}U$$

Then a complete mathematical model can be established:(27)$$B^{s}=K_{M,sensitivity}C_{M,nonorth}C_{b}^{s^{\prime }}B_{e}^{b}+B_{0}^{s}$$

The parameter can be obtained,
(28)$$\begin{array}{l} C_{s}^{s^{\prime }}={\left(C_{s^{\prime }}^{s}\right)}^{-1}={\left(K_{M,sensitivity}C_{M,nonorth}\right)}^{-1}=\left[\begin{array}{ccc} 1 & \sin\delta_{M,1}^{\prime } & \sin\delta_{\prime }^{M,2}\cos\delta_{\prime }^{M,3}\\ 0 & \cos\delta_{\prime }^{M,1} & \sin\delta_{\prime }^{M,2}\sin{\delta^{\prime }}_{M,3}\\ 0 & 0 & \cos{\delta^{\prime }}_{M,2} \end{array}\right]\left[\begin{array}{ccc} 1/{k^{\prime }}_{M,x} & 0 & 0\\ 0 & 1/{k^{\prime }}_{M,y} & 0\\ 0 & 0 & 1/{k^{\prime }}_{M,z} \end{array}\right]\\ =\left[\begin{array}{ccc} 1/{k^{\prime }}_{M,x} & \sin{\delta^{\prime }}_{M,1}/{k^{\prime }}_{M,y} & \sin{\delta^{\prime }}_{M,2}\cos{\delta^{\prime }}_{M,3}/{k^{\prime }}_{M,z}\\ 0 & \cos{\delta^{\prime }}_{M,1}/{k^{\prime }}_{M,y} & \sin{\delta^{\prime }}_{M,2}\sin{\delta^{\prime }}_{M,3}/{k^{\prime }}_{M,z}\\ 0 & 0 & \cos{\delta^{\prime }}_{M,2}/{k^{\prime }}_{M,z} \end{array}\right] \end{array}$$

Therefore, equivalent sensitivity and equivalent non-orthogonal angles are:$$\begin{array}{l} {k^{\prime }}_{M,x}=1/C_{s}^{s^{\prime }}(1,1)\\ {\delta^{\prime }}_{M,1}=\arctan\left[C_{s}^{s^{\prime }}(1,2)/C_{s}^{s^{\prime }}(2,2)\right]\\ {k^{\prime }}_{M,y}=\sin{\delta^{\prime }}_{M,1}/C_{s}^{s^{\prime }}(1,2)\\ {\delta^{\prime }}_{M,3}=\arctan\left[C_{s}^{s^{\prime }}(2,3)/C_{s}^{s^{\prime }}(1,3)\right]\\ {\delta^{\prime }}_{M,2}=\arctan\frac{C_{s}^{s^{\prime }}(1,3)}{\cos{\delta^{\prime }}_{M,3}C_{s}^{s^{\prime }}(3,3)}\\ {k^{\prime }}_{M,z}=\sin{\delta^{\prime }}_{M,2}\cos{\delta^{\prime }}_{M,3}/C_{s}^{s^{\prime }}(1,3) \end{array}$$

### 3.2. Second-Stage Calibration

#### 3.2.1. Three-Position Method

Although the geomagnetic field is a global long-term changing magnetic field, it changes slowly and is almost constant. Therefore, there is a constraint relationship of the measured values when the measurement system is in different directions. According to this theory, the projectile is placed on a non-magnetic platform, the X-axis, Y-axis and Z-axis of the projectile coordinate system point to the north, up and east, respectively. Record the measured value at the initial position as *B_1_^s^*^’^, rotate 180° around the other two axes in the projectile coordinate system and record the measured value as *B_2_^s^*^’^ and *B_3_^s^*^’^. There are six rotation sequence schemes as shown in Table 1.

Considering the ease of operation, without loss of generality, plan 3 is used in the following description, the sequence of rotation is rotating 180° around the Z-axis after the initial position, then rotating 180° around X-axis, illustrated by Figure 4.

Therefore, according to the rotation sequence: the three-dimensional vector of the geomagnetic information in the calibration coordinate system is *B^c^* = [*B_x_^c^*, *B_y_^c^*, *B_z_^c^*]*^T^*; the three-dimensional projection vector of the geomagnetic information in the sensitive orthogonal coordinate system (‘s′’ system) is *B^s^*^’^ = [*B_x_^s^*^’^, *B_y_^s^*^’^, *B_z_^s^*^’^]*^T^*. There is:(29)$$B_{i}^{s^{\prime }}=C_{b}^{s^{\prime }}B_{i}^{b};(i=1,2,3)$$
where, ***B***_1_^*b*^ = [*B_x_^c^, B_y_^c^, B_z_^c^*]^*T*^, ***B***_2_^*b*^ = [*−B_x_^c^, −B_y_^c^, B_z_^c^*]^*T*^, ***B***_3_^*b*^ = [*−B_x_^c^, B_y_^c^, −B_z_^c^*]^*T*^.

Determine $B_{ij}^{s^{\prime }}=\left(B_{i}^{s^{\prime }}+B_{j}^{s^{\prime }}\right)/2$, *i* and *j* are the number 1, 2 and 3 of the positions.

Using the Newton–Raphson method, suppose the $X=[\alpha_{x},\alpha_{y},\alpha_{z},B_{x}^{c},B_{y}^{c},B_{z}^{c}]$, define the initial value $X_{k}$, Taylor expands $X_{i}\left(i=1,2,3\ldots \right)$ at $X_{k}$, omit high-order terms with small values.
(30)ΔZ=J⋅ΔX
where:


$\Delta X=X-X_{k}$



$\Delta Z=\left[H_{k,ij}^{c}-f_{k,ij}(X_{k})\right]$


$f_{k,ij}$ represents the constraint relationship of $\left[\alpha_{x},\alpha_{y},\alpha_{z},B_{x}^{c},B_{y}^{c},B_{z}^{c}\right]$ and ***B^b^***. *J* is the Jacobian matrix of the nonlinear equations:(31)$$J_{9\times 6}=\left[\partial f_{k,12}/\partial X\right]{|}_{X_{k}}=\left[\begin{array}{cc} {\left(J_{12,\alpha ,\beta ,\gamma }\right)}_{3\times 3} & {\left(J_{12,B^{c}}\right)}_{3\times 3}\\ {\left(J_{23,\alpha ,\beta ,\gamma }\right)}_{3\times 3} & {\left(J_{12,B^{c}}\right)}_{3\times 3}\\ {\left(J_{13,\alpha ,\beta ,\gamma }\right)}_{3\times 3} & {\left(J_{12,B^{c}}\right)}_{3\times 3} \end{array}\right]$$
(32)$$\begin{array}{l} {\left(J_{12,\alpha ,}\right)}_{3\times 3}=\left[\partial f/\partial \alpha \right]\\ =\left[\begin{array}{ccc} 0 & -B_{z}^{c}(\cos\alpha_{y}\cos\alpha_{z}) & B_{z}^{c}(\sin\alpha_{y}\sin\alpha_{z})\\ B_{z}^{c}(\cos\alpha_{x}\cos\alpha_{y}-\sin\alpha_{x}\sin\alpha_{y}\sin\alpha_{z}) & B_{z}^{c}(\cos\alpha_{x}\cos\alpha_{y}\sin\alpha_{z}-\sin\alpha_{x}\sin\alpha_{y}) & B_{z}^{c}(\cos\alpha_{x}\sin\alpha_{y}\cos\alpha_{z})\\ -B_{z}^{c}(\sin\alpha_{x}\cos\alpha_{y}+\cos\alpha_{x}\sin\alpha_{y}\sin\alpha_{z}) & -B_{z}^{c}(\cos\alpha_{x}\sin\alpha_{y}+\sin\alpha_{x}\cos\alpha_{y}\sin\alpha_{z}) & -B_{z}^{c}(\sin\alpha_{x}\sin\alpha_{y}\cos\alpha_{z}) \end{array}\right] \end{array}$$
(33)$$\begin{array}{l} {\left(J_{12,\alpha }\right)}_{3\times 3}=\left[\partial f/\partial B^{c}\right]\\ =\left[\begin{array}{ccc} 0 & 0 & -\sin\alpha_{y}\cos\alpha_{z}\\ 0 & 0 & \sin\alpha_{x}\cos\alpha_{y}+\cos\alpha_{x}\sin\alpha_{y}\sin\alpha_{z}\\ 0 & 0 & \cos\alpha_{x}\cos\alpha_{y}-\sin\alpha_{x}\sin\alpha_{y}\sin\alpha_{z} \end{array}\right] \end{array}$$
(34)$$\begin{array}{l} {\left(J_{23,\alpha }\right)}_{3\times 3}=\left[\partial f/\partial \alpha \right]\\ =\left[\begin{array}{ccc} 0 & B_{x}^{c}(\sin\alpha_{y}\cos\alpha_{z}) & B_{x}^{c}(\cos\alpha_{y}\sin\alpha_{z})\\ -B_{x}^{c}(\cos\alpha_{x}\sin\alpha_{y}+\sin\alpha_{x}\cos\alpha_{y}\sin\alpha_{z}) & -B_{x}^{c}(\sin\alpha_{x}\cos\alpha_{y}+\cos\alpha_{x}\sin\alpha_{y}\sin\alpha_{z}) & B_{x}^{c}(\cos\alpha_{x}\cos\alpha_{y}\cos\alpha_{z})\\ B_{x}^{c}(\sin\alpha_{x}\sin\alpha_{y}-\cos\alpha_{x}\cos\alpha_{y}\sin\alpha_{z}) & B_{x}^{c}(\sin\alpha_{x}\sin\alpha_{y}\sin\alpha_{z}-\cos\alpha_{x}\cos\alpha_{y}) & -B_{x}^{c}(\sin\alpha_{x}\cos\alpha_{y}\cos\alpha_{z}) \end{array}\right] \end{array}$$
(35)$$\begin{array}{l} {\left(J_{23,\alpha }\right)}_{3\times 3}=\left[\partial f/\partial B^{c}\right]\\ =\left[\begin{array}{ccc} -\cos\alpha_{y}\cos\alpha_{z} & 0 & 0\\ \cos\alpha_{x}\sin\alpha_{y}\sin\alpha_{z}-\sin\alpha_{x}\cos\alpha_{y} & 0 & 0\\ -(\cos\alpha_{x}\sin\alpha_{y}+\sin\alpha_{x}\cos\alpha_{y}\sin\alpha_{z}) & 0 & 0 \end{array}\right] \end{array}$$
(36)$${\left(J_{13,\alpha }\right)}_{3\times 3}=\left[\partial f/\partial \alpha \right]=\left[\begin{array}{ccc} 0 & 0 & B_{y}^{c}\cos\alpha_{z}\\ -B_{y}^{c}(\sin\alpha_{x}\cos\alpha_{z}) & 0 & -B_{y}^{c}(\cos\alpha_{x}\sin\alpha_{z})\\ -B_{y}^{c}(\cos\alpha_{x}\cos\alpha_{z}) & 0 & B_{y}^{c}(\sin\alpha_{x}\sin\alpha_{z}) \end{array}\right]$$
(37)$$\begin{array}{l} {\left(J_{13,\alpha }\right)}_{3\times 3}=\left[\partial f/\partial B^{c}\right]\\ =\left[\begin{array}{ccc} 0 & \sin\alpha_{z} & 0\\ 0 & \cos\alpha_{x}\cos\alpha_{z} & 0\\ 0 & -\sin\alpha_{x}\cos\alpha_{z} & 0 \end{array}\right] \end{array}$$

Using the least-squares method, $\Delta X=(N^{T}N{)}^{-1}N^{T}\Delta M$ can be obtained, and there is an iterative calculation $X_{k+1}=X_{k}+\Delta X$ until the required accuracy threshold. When using the Newton–Raphson method, its convergence requires the initial value to be selected appropriately. Define the initial value:(38)$$\begin{array}{l} B_{x}^{c}\approx B_{x,23}^{b}\\ B_{y}^{c}\approx B_{y,13}^{b}\\ B_{z}^{c}\approx B_{z,12}^{b}\\ \alpha_{x}=\frac{B_{z}^{c}B_{y,12}^{b}-B_{y}^{c}B_{z,13}^{b}}{{\left(B_{z}^{c}\right)}^{2}+{\left(B_{y}^{c}\right)}^{2}}\\ \alpha_{y}=\frac{B_{x}^{c}B_{z,23}^{b}-B_{z}^{c}B_{x,12}^{b}}{{\left(B_{z}^{c}\right)}^{2}+{\left(B_{x}^{c}\right)}^{2}}\\ \alpha_{z}=\frac{B_{y}^{c}B_{x,13}^{b}-B_{x}^{c}B_{x,23}^{b}}{{\left(B_{x}^{c}\right)}^{2}+{\left(B_{y}^{c}\right)}^{2}} \end{array}$$

#### 3.2.2. Error Analysis

Suppose the measurement error conforms to the normal distribution, for which the mean value is zero and the three-axis magnetic measurement values are independent of each other, satisfies Δ*H_i_^s^*^’^ ~ *N*(0, *σ*^2^), as:(39)$$\begin{array}{l} E\left[\Delta B_{i}^{s^{\prime }}\right]=0\\ E\left[{\left(\Delta B_{i}^{s^{\prime }}\right)}^{2}\right]=\sigma^{2} \end{array}$$

Then,
(40)$$\begin{array}{l} E\left[\Delta B_{k,ij}^{m}\right]=E\left[\left(\Delta B_{k,i}^{m}+\Delta B_{k,j}^{m}\right)/2\right]=0\\ i,j=1,2,3;k=x,y,z\\ E\left[{\left(\Delta B_{k,ij}^{m}\right)}^{2}\right]=E\left[{\left(\Delta B_{k,i}^{m}+\Delta B_{k,j}^{m}\right)}^{2}/4\right]=\sigma^{2}/2\\ \left(i,j=1,2,3;k=x,y,z\right) \end{array}$$

Then,
(41)$$\begin{array}{l} E\left[\Delta Z\right]=0_{9\times 1}\\ E\left[\Delta Z\cdot \Delta Z^{T}\right]=\frac{\sigma^{2}}{2}I_{9\times 9}; \end{array}$$

According to $\Delta X=(N^{T}N{)}^{-1}N^{T}\Delta M$, the statistics of ***X*** satisfy the equation
(42)$$\begin{array}{ll} E\left[\Delta X\right] & =E\left[{\left(N^{T}N\right)}^{-1}N\cdot \Delta M\right]={\left(N^{T}N\right)}^{-1}A\cdot E\left[\Delta M\right]=0_{6\times 1}\\ & E\left[\Delta X\cdot \Delta X^{T}\right]\\ & =E\left[{\left(N^{T}N\right)}^{-1}N^{T}\cdot \Delta M\cdot \Delta M^{T}\cdot N{\left(N^{T}N\right)}^{-1}\right]\\ & ={\left(N^{T}N\right)}^{-1}N^{T}\cdot E\left[\Delta M\cdot \Delta M^{T}\right]\cdot N{\left(N^{T}N\right)}^{-1}\\ & =\frac{\sigma^{2}}{2}{\left(N^{T}N\right)}^{-1}N^{T}N{\left(N^{T}N\right)}^{-1}\\ & =\frac{\sigma^{2}}{2}{\left(N^{T}N\right)}^{-1} \end{array}$$

When *α_x_*, *α_y_*, *α_z_* are small values, the quadratic term and other higher-order terms in the Taylor expansion can be ignored, and the covariance matrix of Δ***X*** can be simplified as:(43)$$\begin{array}{ll} E\left[\Delta X\cdot \Delta X^{T}\right] & =\frac{\sigma^{2}}{2}{\left(N^{T}N\right)}^{-1}\\ & =\frac{\sigma^{2}}{2}\left[\begin{array}{cc} N_{11} & 0_{3\times 3}\\ 0_{3\times 3} & I_{3\times 3} \end{array}\right] \end{array}$$
where $N_{11}=diag\left[\frac{1}{{\left(B_{y}^{c}\right)}^{2}+{\left(B_{z}^{c}\right)}^{2}};\frac{1}{{\left(B_{x}^{c}\right)}^{2}+{\left(B_{z}^{c}\right)}^{2}};\frac{1}{{\left(B_{x}^{c}\right)}^{2}+{\left(B_{y}^{c}\right)}^{2}}\right]$

Therefore, it can be seen that the calibration accuracy of the misalignment angles of the magnetic field information is related to the measurement accuracy of the three-axis magnetic sensor, the magnetic field vector and the projection component of ***H^c^*** in the calibration coordinate system. In order to estimate *α_x_*, *α_y_* and *α_z_* with equal accuracy, it is necessary to project the magnetic field vector ***B***^c^ to the calibration coordinate system with equal components. Determine the intensity of the geomagnetic field as *||**B**||*; it needs to meet the following equation:(44)$$\left\|B_{x}^{c}\right\|=\left\|B_{y}^{c}\right\|=\left\|B_{z}^{c}\right\|=\left\|B\right\|/\sqrt{3}$$

The angle between the magnetic field vector of the calibration site and each axis in the calibration coordinate system is $\pm arc\cos\left(1/\sqrt{3}\right)=\pm 54.73{}^{\circ}$, as:(45)$$\begin{array}{l} E\left[\Delta \alpha_{i}^{2}\right]=\frac{3\sigma^{2}}{4{\left\|B\right\|}^{2}};\\ E\left[{\left(\Delta B_{i}^{b}\right)}^{2}\right]=\frac{\sigma^{2}}{2} \end{array}$$

## 4. Simulation

In order to verify the feasibility of this scheme, suppose the intensity of geomagnetic field is 60,000 nT, add necessary hard magnetism and soft magnetism; the simulation data is shown in Table 2.

Fitting the data as an ellipsoid surface is shown in Figure 5.

According to the analysis above, the parameters are as shown in Table 3.

Compensate the measured value and the fitting diagram as shown in Figure 6.

It can be seen from Figure 6 that the fitting graph of the measured value after compensation is close to a sphere, which is a proper graph under ideal conditions, which proves that this method can eliminate the influence of the deviation of parameters of sensor and other magnetic fields.

Suppose the misalignment angles are [−1°, 2°, 3°], the three-axis components in the calibration coordinate system are [35,468, 35,468, 35,468] nT, as shown in Table 4, to verify the feasibility of the TPC method.

To obtain the optimal results, the available methods are the Las Vegas method and the Monte Carlo method. Because of the limited number of points in the actual sampling, it does not meet the applicated conditions of the Las Vegas method, therefore the distribution of misalignment angles is judged by the Monte Carlo method 10,000 times. The results are shown in Table 5 and Figure 7.

It can be seen from Table 5, Figure 7, in the results of the calibration algorithm based on the TPC method, that the errors of the misalignment angle and the magnetic field component all meet the characteristics of normal distribution, their average values are zero and the three misalignment angles are estimated with equal accuracy when the components are equal. The standard deviation of components is close to the theoretical value 70.7107 nT, it is consistent with the analysis in Section 3 of this paper.

## 5. Experiment and Discussion

In order to verify the scheme described in the article, HMC1053 is the magnetometer in this experiment. The magnetoresistive sensor (AMR) of Honeywell of the United States represents the highest level in the industry; the miniature three-axis magnetoresistive sensor HMC1053 has the characteristics of small size, low power consumption and fast dynamic response. It is widely used in the navigation test of unmanned aerial vehicles, unmanned vehicles and various small smart carriers; as well, it has been used in guided munitions of France, and the “CORECT” combined measurement module for rocket ballistic correction jointly developed by Switzerland and Germany. In this experiment, the sensor is installed on the board by welding; it is important to install the measurement system on a piece of equipment made of metal materials as shown in Figure 8, FVM400 fluxgate is used to measure an appropriate area where the geomagnetic field is stable for calibration. The accuracy of the device is 1 nT. 

Place the system at a certain position and rotate it around the X-axis, Y-axis and Z-axis with multiple attitudes. Without loss of generality, the selected method is to rotate 360° on the vertical plane when the system is moving 45° on the horizontal plane. Obtain measured data of multiple postures, obtain the modulus as shown in Figure 9.

Figure 9 shows that, before calibration, the peak-to-peak variation range of the magnetic field modulus is about 44,090 nT. The ellipsoid fitting graph is as shown in Figure 10.

Equivalent zero offset, equivalent sensitivity and equivalent non-orthogonal angles are shown in Table 6.

The modulus of measured value after compensation in Figure 11.

Figure 11 shows that after the first stage calibration, the peak-to-peak variation range of the magnetic field modulus is about 2200 nT. Compared with the modulus before the first stage, the error reduces by about 20 times.

Then, to calibrate misalignment angles with TPC method, a straight line in the plane of a plastic plank is selected as the X-axis of the calibration coordinate system, the straight line perpendicular to the X-axis in this plane is the Z-axis and the calibration coordinate system is determined according to the right-hand rule. Using FVM400 to measure the magnetic field components and place the plate in the proper position so that the components in the calibration coordinate system are approximately equal, the components are 35,977 nT, 35,871 nT and 36,100 nT.

First, place the projectile on the calibration plane so that the projectile coordinate system is consistent with the calibrated coordinate system, marked as position 1, as shown in Figure 12.

Then, rotate 180° around the Z-axis of the projectile coordinate system, marked as position 2 as shown in Figure 13.

Then, rotate 180° around the X-axis of projectile coordinate system, marked as position 3 as shown in Figure 14. Record the measured value of the three positions in Table 6.

It can be seen from Table 7 before compensating for misalignment errors, the error of the single axis is thousands of nT and, due to the existence of the misalignment angles, the measured values of the X-axis and the Z-axis have also changed when rotating around the X-axis and Z-axis; in theory, they should not change. According to the calibration algorithm elaborated in Section 3, the misalignment angles are [−2.7903°, −3.2721°, 5.0245°] and the compensated values are obtained as shown in Table 8.

The results show that the TPC method can calibrate misalignment angles, compensate measured value according to analysis in Section 2, the error after calibration is within 150 nT and the maximal error is reduced from 6028 nT to 144 nT. The calibration improves the accuracy of the measured value significantly, which indicates that the misalignment error is eliminated effectively. The results of the experiment show that the proposed scheme has high precision, and the complete process can be finished in a few minutes, which has high operability and applicability, making it suitable for calibrating magnetic measurement systems on guided munition on site.

## 6. Conclusions

In this paper, a TSC scheme without additional reference to calibrate tri-axial magnetic measurement system on guide munition is presented. This scheme solves the problem that the projectile coordinate system is not parallel to the coordinate system of the magnetic measurement system, it is suitable for the calibration of the measurement system of various types of projectiles on site without other devices or references.

The simulation and semi-physical experiment verified that the error of the sensor can be eliminated effectively by the first stage and misalignment angles can be calibrated by the second stage. After the complete scheme is applied, the deviation of the measured value has reduced nearly two orders, accuracy of the magnetic measurement system meets the requirements for estimation of the attitude of guided munitions.

## Figures and Tables

**Figure 1 sensors-21-05799-f001:**
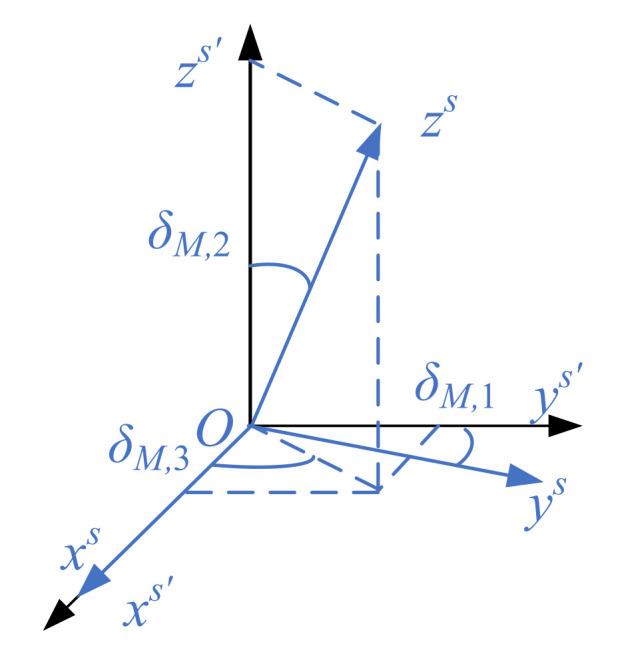
Schematic diagram of non-orthogonal angle.

**Figure 2 sensors-21-05799-f002:**
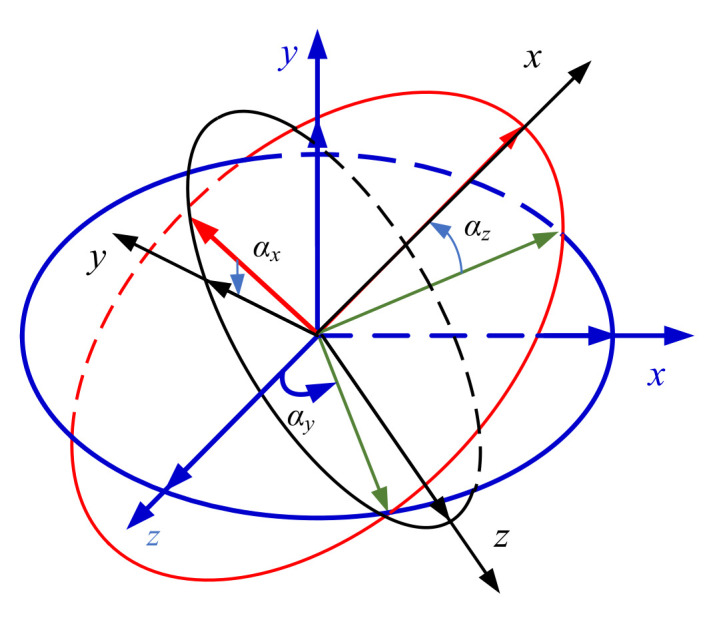
Schematic diagram of installation error angle.

**Figure 3 sensors-21-05799-f003:**
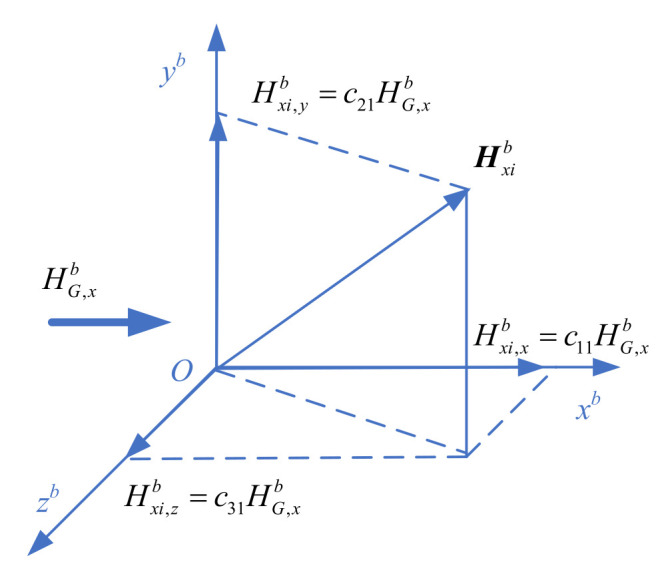
Schematic diagram of decomposition.

**Figure 4 sensors-21-05799-f004:**
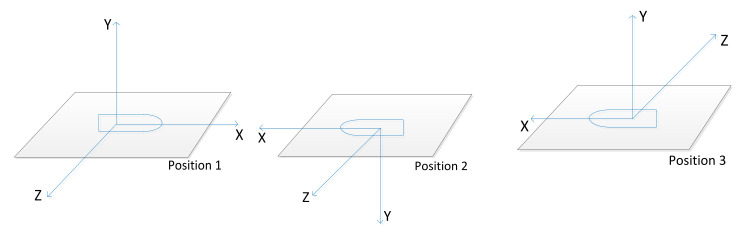
Diagram of rotation sequence.

**Figure 5 sensors-21-05799-f005:**
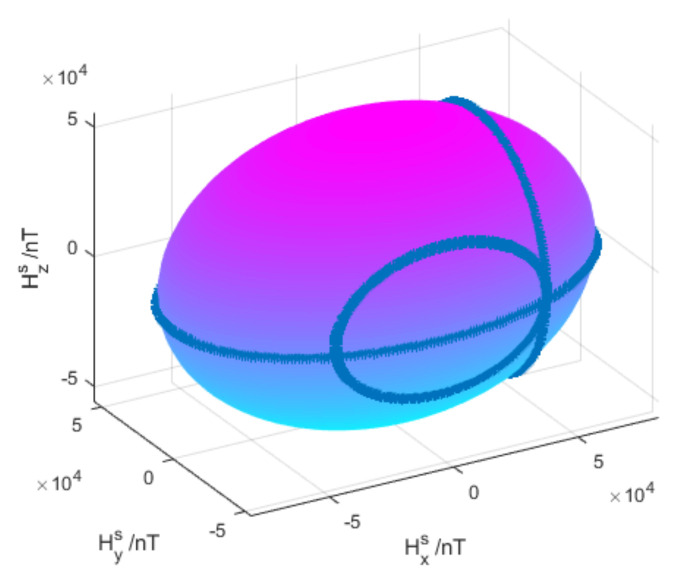
Diagram of ellipsoid fitting.

**Figure 6 sensors-21-05799-f006:**
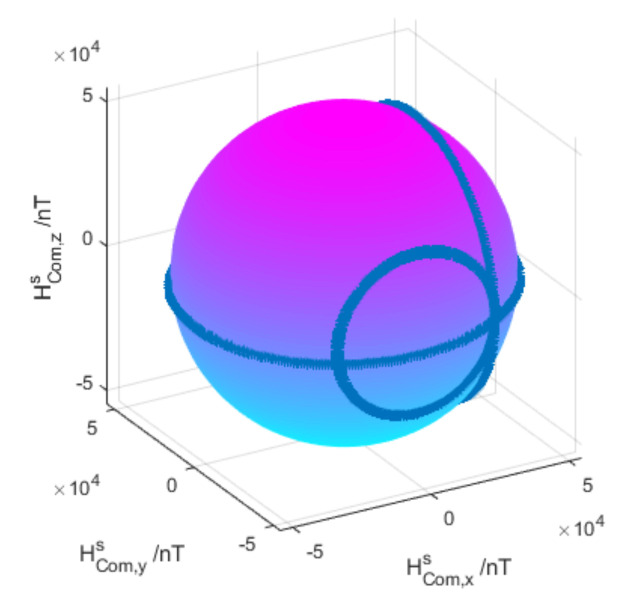
Diagram of data after compensation.

**Figure 7 sensors-21-05799-f007:**
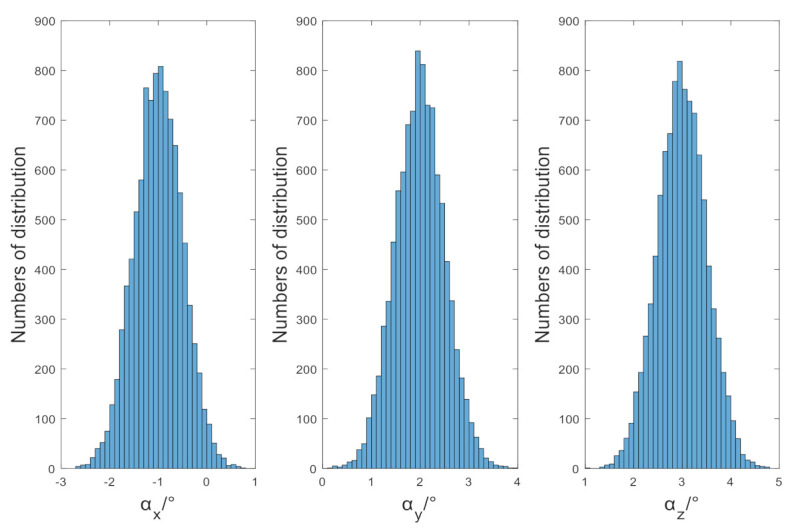
Simulation results of misalignment angles.

**Figure 8 sensors-21-05799-f008:**
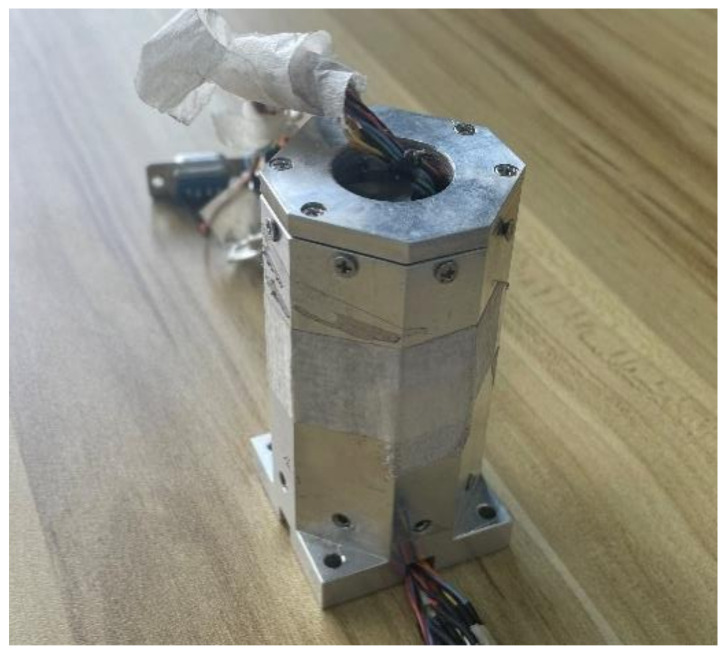
Magnetic measurement system.

**Figure 9 sensors-21-05799-f009:**
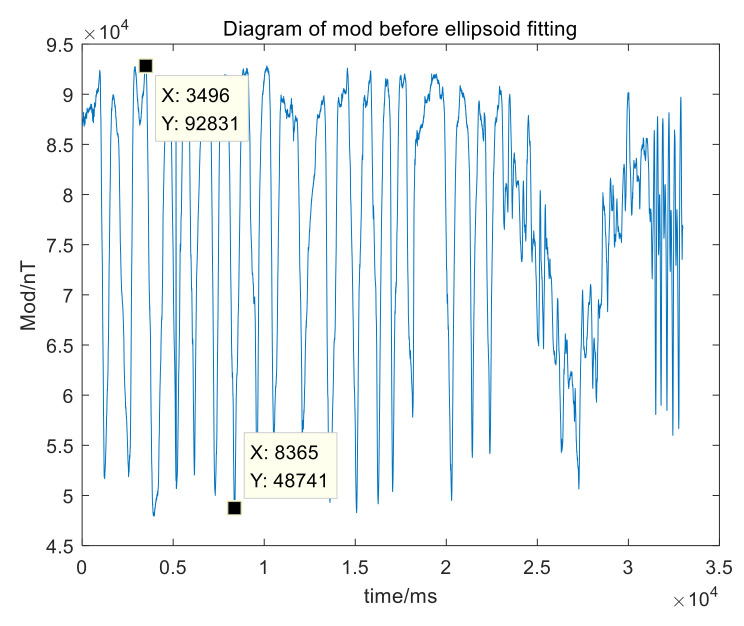
Diagram of modules before calibration.

**Figure 10 sensors-21-05799-f010:**
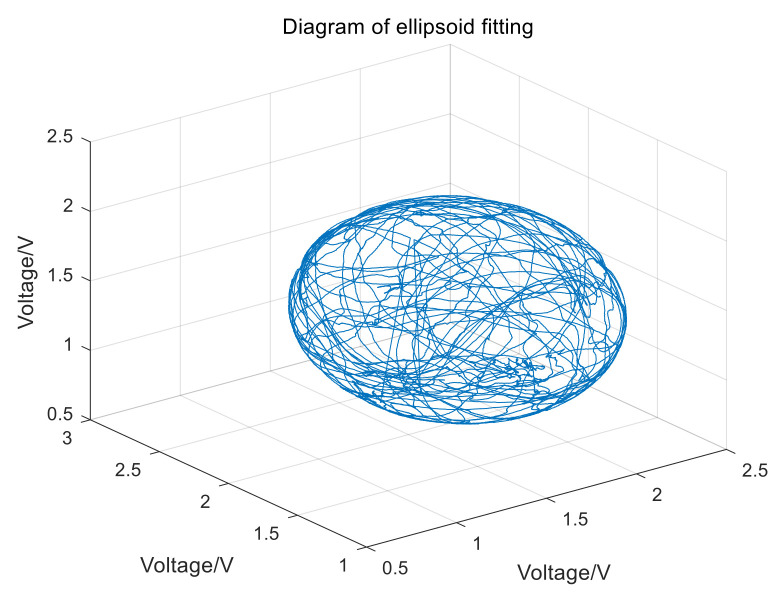
Ellipsoid fitting graph of magnetometer.

**Figure 11 sensors-21-05799-f011:**
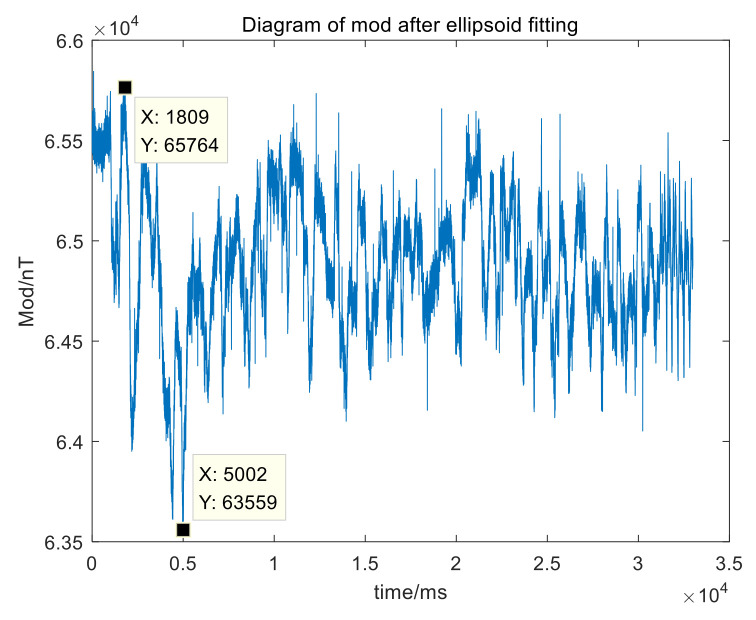
Diagram of modules after compensation.

**Figure 12 sensors-21-05799-f012:**
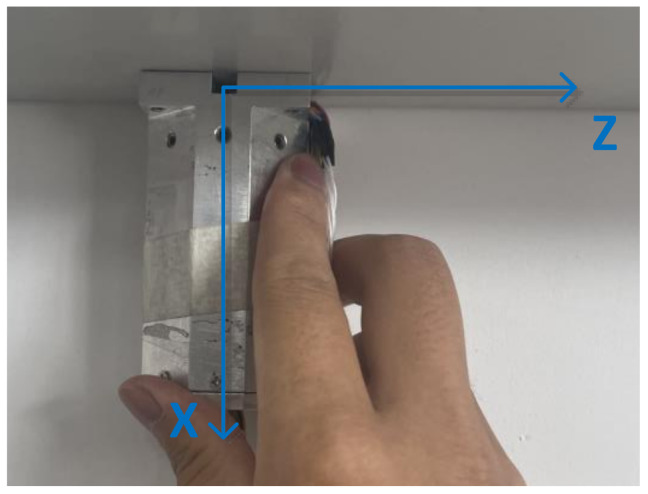
Position 1 in the TPC method.

**Figure 13 sensors-21-05799-f013:**
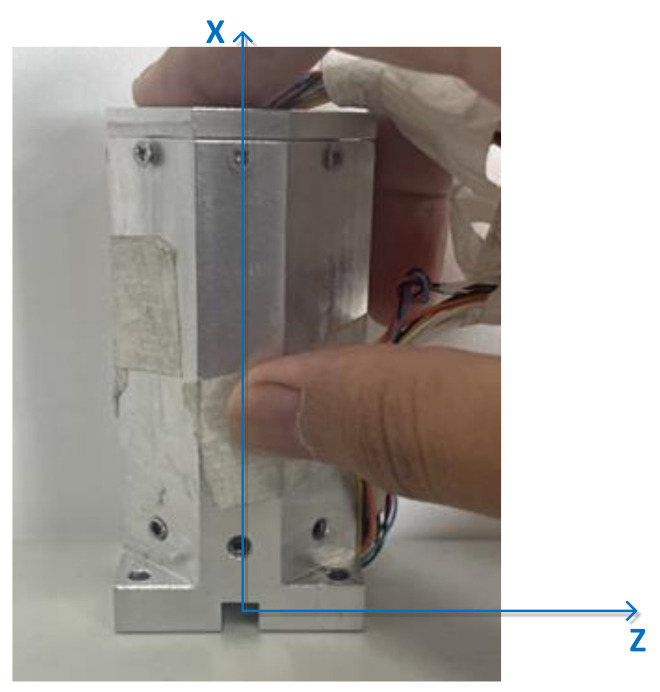
Position 2 in the TPC method.

**Figure 14 sensors-21-05799-f014:**
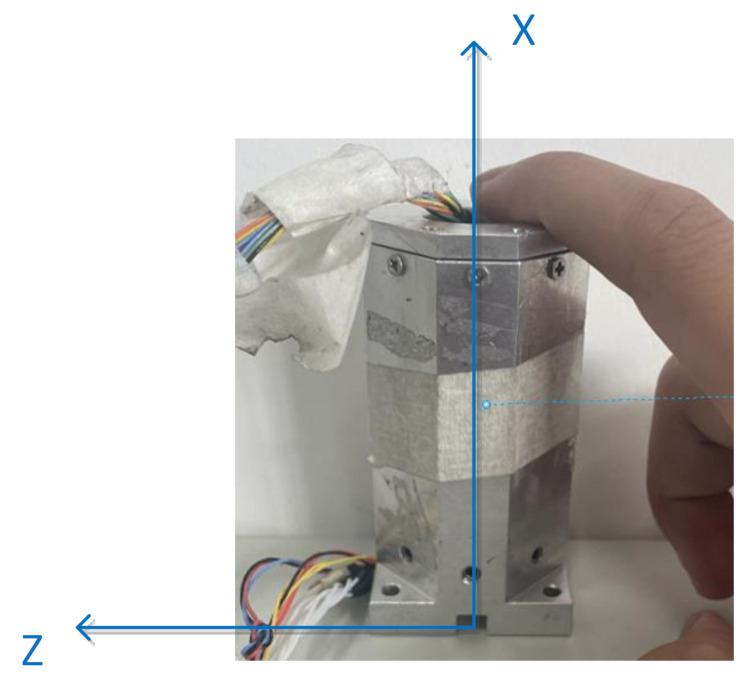
Position 3 in the TPC method.

**Table 1 sensors-21-05799-t001:** Rotation sequence.

	Position 1	Position 2	Position 3
Plan 1	Coincide with the calibration coordinate system	Rotate 180° around *X^b^*	Rotate 180° around *Y^b^*
Plan 2	Coincide with the calibration coordinate system	Rotate 180° around *Y^b^*	Rotate 180° around *Z^b^*
Plan 3	Coincide with the calibration coordinate system	Rotate 180° around *Z^b^*	Rotate 180° around *X^b^*
Plan 4	Coincide with the calibration coordinate system	Rotate 180° around *X^b^*	Rotate 180° around *Z^b^*
Plan 5	Coincide with the calibration coordinate system	Rotate 180° around *Y^b^*	Rotate 180° around *X^b^*
Plan 6	Coincide with the calibration coordinate system	Rotate 180° around *Z^b^*	Rotate 180° around *Y^b^*

**Table 2 sensors-21-05799-t002:** Simulation data.

Hard Magnetism (nT)	Soft Magnetism (nT)
[500, 300, −300]	[30, 100, 100]

**Table 3 sensors-21-05799-t003:** Parameters.

Zero Offset (nT)	Sensitivity	Non-Orthogoanal Angle (°)
[478, 325, −296]	[1.0025, 0.9975, 1.0020]	[1, 1, 2]

**Table 4 sensors-21-05799-t004:** Simulation data.

	Misaligment Angles (°)	Components (nT)
Data	[−1°, 2°, 3°]	[35,468, 35,468, 35,468]

**Table 5 sensors-21-05799-t005:** Simulation results.

	Misalignment Angle (°)			Components (nT)		
	Average Value	Average Error	Standard Deviation	Average Value	Average Error	Standard Deviation
Data	−0.9949	0.0051	0.0828	35,468	0.7335	70.3835
	2.0003	0.0003	0.0821	35,469	1.2859	70.4160
	3.0016	0.0016	0.0825	35,470	2.4139	71.5294

**Table 6 sensors-21-05799-t006:** Results of ellipsoid fitting.

	Zero Offset (V)	Sensitivity (V/nT)	Non-Orthogonal Angle (°)
X	0.1318	0.00001342	−0.9777
Y	0.2660	0.00001334	−7.0315
Z	0.2041	0.00001351	−4.4631

**Table 7 sensors-21-05799-t007:** Measured value before compensation.

	Position 1 (nT)	Position 2 (nT)	Position 3 (nT)
Measured value	[33,529, 30,722, 42,128]	[−34,500, −39,265, 30,148]	[−34,271, 39,764, −31,076]
Deviation	[−2448, −5149, 6028]	[1477, −3394, −5952]	[1706, 3893, 5024]

**Table 8 sensors-21-05799-t008:** Compensated value.

	Position 1 (nT)	Position 2 (nT)	Position 3 (nT)
Compensated value	[35,901, 35,932, 36,057]	[−35,892, −35,934, 36,175]	[−35,937, 35,972, −36,244]
Deviation	[−76, 61, −43]	[85, −63, 75]	[40, 101, −144]

## Data Availability

All data, and related codes of the calibration scheme will be sent to the e-mail of the corresponding author upon request, and appropriate reasons will be provided.

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
