# Peer review of "Two-Stage Calibration Scheme for Magnetic Measurement System on Guided Munition"

_sensors, 2021, doi:10.3390/s21175799_

Round 1

Reviewer 1 Report

This manuscript  discuss about calibration scheme for magnetic measurement system. This paper is write well, however as I reviewer, I have several suggestion:

1) The experimental part must be described in more detail. 

2)  Line 404 : The figure shows that after first stage calibration, - which figure?

3) Many editorial errors should be corrected. 

4) Please check the correctness of equation 28. 

Author Response

Dear editor:

We are grateful for your work on our manuscript, and we have revised the manuscript based on your recommendations.

Point 1: The experimental part must bedescribed in more detail.

Response to point 1:We have added a detailed description of the experimental part with more explanations and illlustrations.

Point 2: Line 404 : The figure shows that after first stage calibration, - which figure?

Response to point 2: We have checked the unclear part of this description and added the number of the figure, now its position is line 421.

Point 3:Many editorial errors should be corrected. 

Response to point 3: We have checked the editorial errors and correct them as we can as possible.

Point 4:Please check the correctness of equation 28. 

Response to point 4:We have correct the equation 28.

We thanks for your patience for our manuscript,and all the revised part have been marked with highlighting. 

Reviewer 2 Report

The paper presents interesting topic of calibration scheme for magnetic measurement system on guided munition. Although the paper is self-explaining and well-written, I have a few recommendations:

  • please re-check the English, there are some grammar mistakes (e.g. line 32 ...system is must be calibrated..., ...errors of measured value[11], which need to rotate the sensor... and others) and some missing articles;
  • I am missing somehow deeper comparison with other types of  calibration methods - e.g. thin shell, spectral analysis and neural networks. I recommend the authors to stress more the differences and benefits of their proposed method.
  • There is missing information why did the authors choose HMC1053 device. Is it already used in some types of munition?
  • Were there tested also different magnetometers of the same type (or different type) with similar results? If yes, please add the results into the paper. It only supports the robustness of the proposed method.

Author Response

Dear editor:

We are grateful for your work on our manuscript, and we have revised the manuscript based on your suggestions,and these parts have been marked with highlighting.

Point 1:Grammar mistakes.

Response to point 1:We checked the grammar and correct the grammar mistakes as we can as possible.

Point 2:Comparison with other calibration methods.

Response to point 2:We added comparison with a method based on BP neural network and alaborate the advantages and disadvantages,and we described the method based on partical swarm optimization in more detail.

Point 3:The reason about why we choose HMC1053 in experiment.

Response to point 3:We have explain this in the experimental part, and cited some examples of military applications.

Thanks for your patience, and all the revised part have been marked with highlighting.